# Intracycle Velocity Variation in Swimming: A Systematic Scoping Review

**DOI:** 10.3390/bioengineering10030308

**Published:** 2023-02-28

**Authors:** Aléxia Fernandes, José Afonso, Francisco Noronha, Bruno Mezêncio, João Paulo Vilas-Boas, Ricardo J. Fernandes

**Affiliations:** 1Centre of Research, Education, Innovation and Intervention in Sport and Porto Biomechanics Laboratory, Faculty of Sport, University of Porto, 4200-450 Porto, Portugal; 2Biomechanics Laboratory, School of Physical Education and Sport, University of São Paulo, São Paulo 05508-030, Brazil

**Keywords:** biomechanics, competitive swimming, performance, velocity fluctuations

## Abstract

Intracycle velocity variation is a swimming relevant research topic, focusing on understanding the interaction between hydrodynamic propulsive and drag forces. We have performed a systematic scoping review to map the main concepts, sources and types of evidence accomplished. Searches were conducted in the PubMed, Scopus and Web of Science databases, as well as the Biomechanics and Medicine in Swimming Symposia Proceedings Book, with manual searches, snowballing citation tracking, and external experts consultation. The eligibility criteria included competitive swimmers’ intracycle velocity variation assessment of any sex, distance, pace, swimming technique and protocol. Studies’ characteristics were summarized and expressed in an evidence gap map, and the risk of bias was judged using RoBANS. A total of 76 studies, corresponding to 68 trials involving 1440 swimmers (55.2 and 34.1% males and females), were included, with only 20 (29.4%) presenting an overall low risk of bias. The front crawl was the most studied swimming technique and intracycle velocity variation was assessed and quantified in several ways, leading to extremely divergent results. Researchers related intracycle velocity variation to coordination, energy cost, fatigue, technical proficiency, velocity, swimming techniques variants and force. Future studies should focus on studying backstroke, breaststroke and butterfly at high intensities, in young, youth and world-class swimmers, as well as in IVV quantification.

## 1. Introduction

Intracycle velocity variation (IVV) is a biomechanical variable that reflects the velocity fluctuation within a swimming cycle and was one of the first swimming-related research topics [1,2] aiming to better understand performance evolution constraints. IVV depends on the interaction between propulsive and resistive forces for each upper limb cycle, with the interaction between these accelerations and decelerations considered an efficiency estimator [3,4]. The first attempt to evaluate this variable was made for the backstroke, breaststroke and front crawl [1], and concluded that common stopwatches could not adequately assess swimming velocity (changes were observed within an s or an m). Velocity was measured with a natograph (recording the distance travelled every 1/5 of an s), and its variation was observed in each studied swimming technique (with front crawl being the fastest due to its smoothness). At that time, swimming was associated with motor cars’ mechanics since, if driving with a variable speed would be wasteful, the same should occur in the human machine. This study provided important insights and investigation lines for the current topic.

Afterwards, the natograph was improved [2,5,6,7], with several mechanical devices beginning to be used (cable speedometers [8,9], accelerometers [10], and other gadgets [11]), all characterized by a mechanical connection to a swimmer’s anatomical point. Despite the incapacity to monitor the swimmer’s bodily inertia due to the constant change in the position of the centre of mass, these methods were very interactive and relevant to training due to the immediate output availability. Cinematography was also very common for evaluating IVV [12,13,14], qualitatively and quantitatively assessing the movements in a three-dimensional nature with (at least) two cameras. These image-based methods, usually involving the digitisation of film or video images, presented similar issues related to the body inertia capture, as well as image distortions, water bobbles and waves, parallax, digitising and calibration errors, and reduced interactivity (due to the delay between data collection and the swimmer feedback as a result of image processing).

Methods dealing with the centre of mass motion have the abovementioned problems but are even more time-consuming and complex. Nowadays, depending on the aims of IVV investigation, researchers are divided between using an anatomical fixed point or the centre of mass [15,16,17]. Considering the accessibility of mechanical methods, the agreement between these measures was evaluated, but the centre of mass reference was constantly overestimated, and it is axiomatically considered a gold standard in those comparisons [17,18,19]. Due to the current approach to this issue, forward hip movements were considered a good estimate of the swimmers’ horizontal velocity and displacement, being relevant for diagnostic purposes but not representing the movement of the centre of mass [15,16,20]. Hip error magnitude should also be considered because it overestimates swimming velocity and, consequently, the IVV of the four conventional swimming techniques [17,18,19].

Despite the above-referenced methodological concerns, the association between swimming IVV and performance continues to be investigated even though the findings are quite divergent. Increases in velocity were associated with lower [3,21], stable [22,23,24,25,26,27,28,29,30,31,32,33] and higher IVV [34,35] in different swimming techniques. Better propulsive continuity in front crawl and lower swimming economy in breaststroke and butterfly (due to elevated resistive forces and amount of work) are the suggested explanations. In addition, when comparing competitive swimming levels for the same pace and swimming technique, better swimmers were observed to have higher [36,37], lower [10,21,23,33,34,38,39] or similar IVV [40,41] values compared to their counterparts. Regarding conventional swimming techniques, breaststroke presents the highest IVV values, followed by butterfly, backstroke and front crawl [3], although alternative techniques’ scores are very similar [42].

Considering the IVV research background and its significance to assess biomechanical development in swimming, the aim of the current study wa to accomplish a systematic scoping review of IVV in competitive swimming regarding the four conventional techniques, assessment and quantification methods, participants’ information (sex, competitive level and age category), protocols, and association with swimming economy and hydrodynamic drag. The closest work to a review about IVV is a book chapter [43] addressing it as a relevant variable to assess swimming biomechanical and coordinative development, as well as its association with swimmers’ technique, exercise intensity, economy and fatigue.

## 2. Materials and Methods

The current systematic scoping review protocol was designed according to PRISMA 2020 [44] and Prisma-ScR guidelines [45], as well as Cochrane recommendations [46]. The protocol was created and pre-registered as an OSF project on 6 July 2022 (https://osf.io/m43pj, accessed on 23 December 2022).

### 2.1. Eligibility Criteria

Original peer-reviewed articles and texts from the Proceedings Book of the Biomechanics and Medicine in Swimming, published in any language or date, were included in the current study. Letters, editorials, meetings abstracts, commentaries, and reviews were excluded. The eligibility criteria were defined by the Population, Exposition, Comparator, Outcomes and Study (PECOS) design model, in accordance with PRISMA guidelines: (i) population (competitive swimmers of any sex, with no injuries, excluding triathletes, divers and Paralympic athletes and artistic and open-water swimmers); (ii) exposure (IVV assessments at any swimming distance, pace, technique and protocol); (iii) comparison (not mandatory if intervention was performed); (iv) outcome (IVV was the primary outcome, with the secondary outcomes being described in the 2.6. data items subsection and not used as inclusion/exclusion criteria) and (v) study design (no limitations for the study strategy).

### 2.2. Information Sources

Searches were conducted until 6 July 2022, in the PubMed, Scopus and Web of Science literature databases, as well as in the Proceedings Books of the Biomechanics and Medicine in Swimming Symposia (no filters were applied). After the automated searches, the reference lists of the included studies were screened and prospective snowballing citation tracking was performed in PubMed, Scopus and Web of Science databases. Two external experts (holding a PhD in Sport Sciences and having considerable published research on the topic) were consulted to provide further suggestions of potentially relevant studies. Included studies’ errata, corrections, corrigenda and retractions were sought [46].

The International Symposia for Biomechanics and Medicine in Swimming have been held every four years since 1970 and are considered the most prestigious international aquatic-oriented scientific congresses. These meetings have provided the swimming science community with some of the most outstanding contributions books and collections (available at https://www.iat.uni-leipzig.de/datenbanken/iks/bms/ accessed on 6 July 2022), as sought and valuable as some of the available studies published in high-impact, peer-reviewed journals. All submissions go through a peer review process, leading to a collection of peer-reviewed scientific papers, serving as a valuable resource for all who are interested in keeping up to date with aquatic research. Relevant pioneering works were published in the 13 editions of the Symposium, adding relevant information to the current review. 

### 2.3. Search Strategy

The general search strategy used free text terms applied to the title or abstracts: swim* AND intracycl* OR “intra-cycl*” OR IVV AND velocity OR speed* OR accelera* OR quick*. The full search strategy for each database is shown in Table 1.

### 2.4. Selection Process

Two authors (AF and JA) independently screened all the database records and performed the manual searches, as well as snowballing citation tracking, with disagreements decided by a third author (RJF). Automated removal of duplicates was performed using EndNote^TM^ 20.3 (Clarivate^TM^, Philadelphia, PA, USA), but manual duplicate removal was required.

### 2.5. Data Collection Process

Two authors (AF and BM) independently collected data, and, in the case of disagreements, a third author (RJF) provided arbitrage. No automation tools were used, and a specifically tailored Excel worksheet was created for the extraction of raw data.

### 2.6. Data Items

The current study’s primary outcome was IVV assessment in the four conventional swimming techniques (according to the above-referred defined eligibility criteria). Velocity assessment methodologies, IVV quantification, participant and protocol information, and associations with swimming economy or hydrodynamic drag were the secondary outcomes. Velocity can be assessed by mechanical, image-based and mixed methods, and IVV can be quantified by the (i) difference between maximal and minimum instantaneous velocity (dv); (ii) ratio of the mean velocity/difference between the maximal and minimum instantaneous velocity; (iii) ratio of the minimum and maximum velocities/intracycle mean velocity (dv/v); (iv) coefficient of variation (CV); and (v) other. 

Regarding participants’ characteristics, we have included studies with samples of female, male or both sexes and young (<14), youth (between 15–16), junior (between 17–18), senior (>19) or master (>25 years) swimmers (following the World of Aquatics stratification). Aiming for a homogeneous classification of competitive level, two authors (AF and JA) applied the Participant Caliber Framework [47] using training volume and performance metrics to classify participants as sedentary, recreational, trained, highly trained, elite and world class. Swimming paces were established according to the intensity training zones, with maximal corresponding to sprint (25–50 m), extreme to anaerobic power (100 m), severe to anaerobic capacity (200 m), heavy to aerobic power (400 m), moderate to aerobic capacity (800 m) and low to prolonged aerobic capacity (>1500 m). Studies were conducted in swimming pool and in swimming flume conditions, and information was gathered regarding the included studies that associated swimming economy or hydrodynamic drag with IVV.

### 2.7. Studies’ Risk of Bias Assessment

Risk of bias in individual studies was judged using Cochrane’s Risk of Bias Assessment for Non-randomized studies (RoBANS; [48]), evaluating six domains: (i) the participant selection; (ii) confounding variables; (iii) the exposure measurement; (iv) the outcome assessments blinding; (v) incomplete outcome data; and (vi) selective outcome reporting.

### 2.8. Effect Measures

IVV mean ± SD or median ± IQR values were calculated, and, when needed, two authors (AF and BM) independently extracted data from graphs using the WebPlotDigitizer v4.5 (Pacifica, CA, USA) [49].

### 2.9. Synthesis Methods

A narrative synthesis of the main findings was performed and supplemented with an interactive evidence gap map (generated by EPPI-Mapper v.2.2.3, London, UK, powered by EPPI Reviewer and created by the Digital Solution Foundry team). This map can be accessed online, providing interactive ways to visualize the current review’s included studies (including authors, abstracts and keywords) and the primary and secondary outcomes.

## 3. Results

The initial search identified 227 potentially relevant articles, with 126 being duplicates, which were consequently removed (Figure 1). Following the titles and abstract screening, 17 and 10 studies were excluded by eligibility criteria and article type (respectively). After the seventy-four full texts were screened, one was excluded by type [50], six by exposure [51,52,53,54,55,56], seven by outcomes [57,58,59,60,61,62,63] and one by participant [64] eligibility criteria. Reference list analysis revealed 31 studies on the topic as potentially meeting the inclusion criteria, with full-text analysis excluding 10 articles by type [65,66,67,68,69,70,71,72,73,74], 2 by exposure [75,76] and 8 by outcomes [18,77,78,79,80,81,82,83]. Seven additional studies from snowballing citation tracking process were deemed eligible for inclusion, and all were included [29,84,85,86,87,88,89]. Expert consultations did not yield any new studies, so the combined total sample was n = 76 corresponding to 68 trials. Studies from the same trial were grouped for the analysis [4,22,25,26,27,29,30,31,33,39,90,91,92,93].

### 3.1. Studies Risk of Bias Assessment

Sixty-eight trials were considered for judging risk of bias, with 20 [6,20,28,36,37,40,41,87,88,89,94,95,96,97,98,99,100,101,102,103] and 48 considered as having overall low and high risk (respectively). The selection of participants showed a low risk of bias for 79% of the trials due to the overall purpose of evaluating competitive swimmers (Figure 2). However, 19% of the trials presented high risk due to the unbalanced number of females versus males [7,104,105], heterogeneity of participants [86,92,106,107], lack of information [10,92,108], or the non-competitive or inexperienced participation in the trials [84,109,110,111]. Two studies [26,27] were judged unclear because of the uncertainty of how swimmers were analysed. Fifty-one percent of the trials had a high risk of bias in the domain of confounding variables due to participant-related problems (lack of information [10,23,26,27,108,109,110,112], swimmers with different characteristics mixed in the same group [15,17,24,85,86,92,104,107,111,113,114,115], swimmers experience [3,84,116,117] and specialty [118]) and protocol-related problems (snorkel use [4,21,25,30,34,35,42,90,91,105,119,120], possible fatigue effect [32,121] and different evaluation conditions [122]). 

Considering that no data were provided concerning the validity and reliability of the software used or whether the process was fully automated in the different studies analysed, exposure measurement was judged unclear for 63% of the trials. High risk was evaluated for 6% of the trials with specific measurement issues; in particular, (i) the electrical resistance variation method had not been previously validated, with authors not providing proof of its reliability [104]; (ii) the preparation procedures and the evaluation protocol were performed for different swimming techniques [123]; (iii) various devices were used for different swimmers, and the evaluation frequency varied substantially in a retrospective study [124], raising questions concerning the actual measurement exposure consistency; and (iv) evaluations did not respect the same time period from the main competitions [125].

Many trials (74%) did not mention outcome assessment blinding and it was unclear if video analysis was fully automated (probably interfering with the measurements). High risk was attributed to 7% of the trials due to no blinding and to the inexistence of data concerning the reliability of the automated process [4,16,25,30,31,90,91,93,116,124]. Due to an absence of information on whether the selected swimmers were part of a larger sample, incomplete outcome data were judged unclear for 88% of the trials, except for a case study [125] and a trial that included an a priori sample-power analysis [109]. High risk was evaluated for 9% of the trials due to missing data, given that this could influence the study outcomes [4,19,25,29,30,39,90,91,111,126,127]. Eighty-eight percent of the trials had no pre-registered protocol to compare to, with the selective outcome reporting unclear. High risk was judged for the trials belonging to the same study [4,22,25,26,27,29,30,31,33,39,90,91,93] and for those that did not fully report the pre-defined primary outcomes [19,104].

### 3.2. Studies Characteristics

The included trials’ main characteristics are presented in Table 2. Across the 68 trials, 1440 swimmers were evaluated for IVV (55.2% male and 10.7% missing information), with n = 1–126 sample sizes and 11.7 ± 0.8–42.5 ± 9.5 years of age. Some trials did not present information regarding IVV [16,19,111,120], female swimmers’ participation [10,23,26,27,92,102,112], competitive level [104,110], age category [7,10,21,104], or protocol intensity [111]. Thirty-nine trials assessed IVV as the main study purpose, of which three analysed and described the swimming cycles curves [7,104,118]; nine related IVV with coordination [22,23,26,27,30,31,32,33,103], six with swimming economy [21,30,34,35,90,105,123], six with fatigue [26,27,84,107,108,112], six with technique [4,36,41,107,111,113] and five with velocity [3,6,37,41,124]; three analysed different swimming techniques variants [35,106,126]; two related to force [94,99]; six were methodological [10,17,19,20,86,87]; one was a dynamical systems approach [40]; and one was a training intervention [88].

IVV was not the primary outcome in 31 trials but was included in a larger analysis, being described [127] and analysed together with anthropometric, kinematic, energetic, coordinative neuromuscular activity and other biomechanical variables [25,30,39,91,97,110,117,119]. Trials also related IVV with coordination [28,29,95], swimming economy [42,92], fatigue [28,29,95,121,128], technique [24,93,120] and velocity [125]. Thus, IVV was included in methodological approaches [15,16,89,96,100], dynamical systems approaches [85,102,114,122] and training interventional trials [29,98,101,109,115]. No conflicting interests were declared or were not addressed by 34 and 66% of the trials. Funding information was not reported by 54% of the trials, while 40% had financial support. Trials dissemination was growing over time (records were published every year) and 2016 was the year with the most publications (seven records). 

### 3.3. Evidence Synthesis

The evaluation of the evidence gap map and trials’ risk of bias can be accessed through the Appendix A. IVV was assessed in 46 front crawl, 10 backstroke, 24 breaststroke and 14 butterfly-related trials, most of them focusing on mixed and male-only groups regarding swimmers’ sex (56 and 39%, respectively). High-level swimmers were the most studied, followed by elite and trained, recreational, world-class and sedentary swimmers (37, 25, 26, 9, 3 and 1%, respectively), from which senior, youth and junior, young and master swimmers participated (50, 18, 18, 10 and 5%, respectively). Regarding the protocol intensity, most trials focused on swimming at sprint and severe intensities (36 and 19%), and fewer implemented incremental protocols that include other intensities (extreme, heavy, moderate and low: 11, 11, 12 and 11%, respectively). Trials conducted in swimming pool conditions were used 99% of the time.

Image- (47 and 53% in two and three dimensions) and mechanical-based methods were used (56 and 41%, respectively), with speedometers being mostly selected (82%). Velocity was calculated using an anatomical fixed point as a reference, most of the time with the hip chosen (and only twice selecting the head/neck) rather than the centre of mass (71 and 29%, respectively). The coefficient of variation was preferred regarding IVV quantification versus the difference between the maximum and minimum instantaneous velocity (dv; 61 and 7%, respectively), the ratio maximum and minimum instantaneous velocity difference/intracycle mean velocity (dv/v; 7%), the ratio of the mean velocity/difference between the maximal and minimum instantaneous velocity (3%) and other methods (such as cycle characterization, curves acceleration and dynamic indexes; 23%). Twenty-five trials reported variables associated with swimming economy (such stroke length and stroke index) and only two reported hydrodynamic drag related variables.

Front-crawl-related trials almost covered all secondary outcomes, even though gaps were identified for the four conventional swimming techniques. No trials were conducted with world-class swimmers focused on extreme, heavy, moderate and low intensities; used accelerometers; or quantified IVV with overall methods. Young swimmers were not used as samples in trials that were conducted at extreme and low swimming intensities, accelerometers were employed, and, when characterizing these age group IVV, its quantification was performed using only three methods. Master swimmers were not called to participate in protocols with extreme intensity and were not evaluated using accelerometers, while IVV quantification in this population was conducted only through the coefficient of variation. Trials using youth/junior, world-class, elite, highly trained and trained swimmers did not have associated IVV and hydrodynamic drag.

### 3.4. Study Results

Higher-level swimmers presented superior mean velocities for the same swimming intensity, but IVV was not related to swimming competitive levels or to the mean velocities regarding the four swimming techniques (Table 3). Except for front crawl, studies were mostly interested in analysing IVV when swimmers were performing at maximal intensity. IVV was not related to mean velocity in front crawl or backstroke [37,40,41,100], even if a non-linear relationship was also observed (with the velocity increase leading to a IVV decrease in young swimmers in the four swimming techniques [3] and in the swimmers with high-level front crawl [4]). Data gathered from so many swimmers and diverse samples should be cautiously analysed. Some outputs were obtained from a single trial performed at a specific swimming intensity, while others were gathered by averaging the data available. In addition, in some studies, swimmers from different competitive levels were pooled, and data were presented as a single group.

In breaststroke, IVV is usually quantified by dv/v (m/s), as presented in Equation (1), with vmax,LL as the maximum centre of mass’s velocity achieved at the end of lower limb propulsion; vmin,LL as the first minimum peak of the centre of mass’s velocity following upper and lower limbs recovery (corresponding to the beginning of lower limb propulsion); vmax,UL as the maximum centre of mass’s velocity at the end of the upper limb propulsion; and vmin,T as the minimum centre of mass’s velocity during the transition between upper and lower limb propulsion (corresponding to the centre of mass’s velocity during gliding).
(1)$$\mathrm{IVV}=\frac{\mathrm{vmax},\mathrm{LL}-\mathrm{vmin},\mathrm{LL}+\mathrm{vmax},\mathrm{UL}-\mathrm{vmax},T}{\mathrm{vmean}}$$

Some trials showed periodic velocity fluctuations related to the upper limbs’ actions and the rate and the number of peaks per cycle, with a higher IVV range in lower- than in higher-level swimmers [7,104,118,127]. Furthermore, successful swimmers were able to more effectively combine intracycle peak velocity with relatively longer cycle periods [6]. When a front crawl technical training intervention period was conducted, IVV decreased [29,88,109] or did not change [98,115]. Although propulsive and drag forces were higher in swimmers of superior level, larger index of coordination values for front crawl were also presented even if IVV did not change across intensities [10,21,23,33,34,38,39], suggesting that better propulsive continuity allows a stable IVV [22,24,25,26,27,28,29,30,31,32,33]. Conversely, IVV increased throughout paces in less skilled swimmers [23]. IVV for highly trained swimmers was lower than for trained counterparts at all front crawl swimming velocities (in both senior and youth age groups) [23,39] but in backstroke, IVV did not differ between elite and highly trained swimmers [40].

IVV was directly related to swimming economy in the four swimming techniques [21,34,35,105,123,126] even though, in one study, no association between these variables was reported [90]. However, front crawl and backstroke IVV did not differ; nonetheless, lower energy cost values for front crawl vs. backstroke were observed [42], and they showed a tendency to decrease in a maximal lactate steady-state test [92]. Similarly, swimmers maintained their IVV values when performing at submaximal intensity, but IVV rose at maximal intensity [84,107,108,112,123,128], even though others described no changes [26,27,121]. This IVV increase with effort is probably justified by the progressive increase in fatigue, resulting in swimmers becoming less mechanically efficient. Swimmers with higher intracycle force variation also presented higher IVV values, leading to a progressive decrease in performance [94,99].

Methodological trials mainly assessed the relationship between the hip and the centre-of-mass kinematics to provide simpler methods to quantify IVV in swimming. It seems consensual that the hip does not adequately represent the centre of mass in intracycle variation in butterfly, breaststroke and front crawl. Some authors clearly state that this anatomic point should not be used in this kind of assessment [16,19,20] because it greatly overestimates the swimmer’s real variation in velocity [15,17]. Other trials aimed to validate methods to quantify and express IVV [10,86,87,89,96]. When applying dynamical system approaches to swimming, nonlinear properties can be observed [114], with their magnitude differing according to the swimming technique and the swimmer’s level. The breaststroke and butterfly techniques displayed more complex (but predictable) patterns [85,114,122] and elite vs. non-elite swimmers’ performances were more unstable and complex (even though their IVV did not differ) [40].

## 4. Discussion

The current systematic scoping review focused on the IVV assessment in swimming that is retrospectively available for almost a century. The IVV-related trials’ main interest is in the interactions between the cyclical propulsive and drag forces, which help understand the cyclic effectiveness of the upper and lower limbs while swimming and, consequently, swimmers’ technical efficiency. In the first studies on IVV, breaststroke was the most studied swimming technique due to the simultaneity between the movements of the upper and lower limbs (which allowed researchers to easily identify when these movements were occurring) [6,118]. Then, new methodologies were developed, with researchers focusing their attention on the four conventional techniques, but our results showed that front crawl aroused greater interest. It is now accepted that the techniques with simultaneous movements (butterfly and breaststroke) present higher IVV than those with alternated movements (front crawl and backstroke) due to the mechanical impulses applied to the swimmer’s body [3,114,122]. Furthermore, the alternated techniques’ IVVs are very similar due to the biomechanical similarities between the front and back crawl (an “old” term used to designate backstroke) [42].

From the analysed trials, we could observe that male swimmers were the most studied even though mixed groups were also used due to the interest in checking differences between female and male swimmers (particularly regarding anthropometric characteristics [39,117], mechanical power output [6,33], technical proficiency and hydrodynamic profile [33,126]). Researchers focused their attention on trained, highly trained and elite swimmers, with the most elevated competitive levels being preferred for analysis. Most trials focused on senior swimmers, displaying strong confidence in results due to their experience. The same was not observed for trials conducted in master swimmers, with considerable gaps found, probably due to their heterogeneity of age and competitive level. Swimmers were mainly evaluated using maximal-intensity protocols to assess the kinematics directly related to the competitive events with the most participation (the 50 and 100 m distances). The 200 m distance was also often investigated, since its metabolic characteristics are important determinants of the kinematic variables’ behaviour during these mixed aerobic–anaerobic events [4,37]. Few studies have focused on the backstroke, breaststroke and butterfly techniques at heavy, severe and extreme intensities.

The included trials used distinct evaluation protocols, with some analysing nonbreathing cycles [15,24,26,27,31,32,85,93,96,104,105,119,120,121,127] and other not reporting the breathing condition or the inclusion of a specific space in which the participants were not allowed to breathe [3,6,10,16,17,19,21,22,23,24,25,30,31,33,36,37,42,85,87,88,90,91,93,95,104,107,108,111,112,115,116,122,124,126,127]. Even though breathing was shown to lead to coordination asymmetry [129], upper-limb-cycle kinematics with individual breathing patterns presented IVV similarities to those in apnoea [41]. Data from trials that used a snorkel for assessing oxygen consumption should be carefully analysed [4,25,30,34,35,42,86,89,90,91,102,105,107,108,119,123]. Concerning the use of the hip vs. the centre of mass for assessing IVV, it was clear that the latter was the most reliable method to measure kinematical variables, although some authors still consider hip movements to provide a good IVV estimate [3,15]. These methods were previously compared with the hypothesis that the hip represented the centre of mass (and not the opposite), which was considered a priori the best methodology [15,16,17,19,20]. Future studies should clarify why the centre of mass is the gold standard considering the complexity of evaluation.

As a consequence of specific front crawl intervention protocols, IVV decreased or remained stable due to better swimming technique [6,102,106,111,113]. This also might have happened in other swimming techniques, with butterfly IVV decreasing when the hands’ velocity at the end of the underwater path and the vertical velocity during the lower limbs’ actions increased, and the velocity during the hands’ entry decreased [111,113]. The hands, trunk and lower limbs role are also fundamental for lowering IVV [4,6,93,126]. Even though it is widely accepted that lower IVV should be achieved for enhanced performance, IVV has no standardized values and is highly variable according to the studied population and the methods used. Therefore, it would be very useful to implement more frequent intervention programs with strategies to upgrade swimmers’ technique and overall performance.

Researchers have started to characterize swimming cycles’ shape and number of peaks, developing quantification methods such as the absolute average velocity, root mean square [10], coefficient of variation and range of maximum and minimum velocities in a cycle [130]. Unfortunately, only one work compared these measurements [131], concluding that the coefficient of variation was the only approach sensitive to the mean swimming velocity and to the instantaneous velocity dispersion during the cycle. Mathematically, it is the more accurate method for IVV quantification but it may overestimate its value in breaststroke (due to this technique’s complexity regarding mechanical impulses and coordination). Nevertheless, even this measure does not reflect the hydrodynamic drag characteristics, and it may be helpful to develop a new method of IVV determination.

Swimmers at a higher level present higher IVV values due to their capacity to generate and sustain the highest velocities (rather than being more economical), displaying larger amplitude of velocity [36,124]. However, breaststrokers eliminated in the preliminaries of a World Swimming Championships displayed higher IVV values than those that qualified for the semi-finals [127], probably as a result of a very low minimal instantaneous velocity (and not necessarily related to the maximal velocity value achieved within a cycle). In short distances, depending on the swimming technique, better swimmers find solutions to improve technical proficiency, producing high mechanical power to generate superior propulsive forces, reducing hydrodynamic drag, and adopting greater propulsive continuity [33,34,38,41], which will cause different IVV.

The quality of the trials included in the current study can be questioned due to the lack of detailed information and uncertainty of the evidence provided (being indeterminate whether it would result in a high or low risk of bias). Disregarding the already mentioned factors that influenced a high risk of bias, most variables were unclear because it the validity and reliability of the exposure measurement were not mentioned, nor were the blinding of the outcome assessment or even the information about whether swimmers belonged to a larger sample. In the scope of swimming, experimental protocols aim to replicate swimmers’ performance and are not usually registered in databases. Furthermore, the current scoping review included trials since 1971 that were not as concerned about the studies’ quality as is dictated today.

## 5. Study Limitations

The number of included trials highlighted the importance and utility of performing a systematic scoping review in swimming IVV. We believe that including the Proceedings Books of the Biomechanics and Medicine in Swimming Symposia strengthened our work, since this book series contains several important documents that added relevant information to the current review. This research aimed to provide an overall representation of the IVV scope of competitive swimming, but we recognize that considering IVV calculations in conditions such as using snorkelling or swimming with/without breathing could affect its interpretation. For sake of the clarity, those studies were properly identified.

## 6. Conclusions

The current study compiles the studies available on the topic of the swimming IVV in the most respected and well-known literature databases. We have described the literature gaps and the most interesting IVV-related topics within almost the past century. IVV was often used in front-crawl-related studies, involving mixed samples and senior swimmers that performed at sprint intensity in swimming pools and were evaluated with cable speedometer using an anatomical fixed point as a reference and that quantified IVV using the coefficient of variation. There is a clear need for investigating backstroke, breaststroke and butterfly swimming techniques performed at heavy, severe and extreme intensities. Since these paces correspond to the characteristics of the official competitive events, it would be imperative to assess them more often. Young and youth swimmers were less studied, even though their performance development in swimming is important in their training process throughout their careers. It would be very helpful to evaluate world-class swimmers as well to acknowledge the top-level performers’ behaviour. Although there is no proof that the coefficient of variation is the best measure to assess IVV, researchers generally agreed that it best reflects the velocity fluctuations in swimming.

## 7. Future Directions

Future investigations should cover the gaps found in the current study to allow for meaningful results and possible comparisons. IVV measurements should be revised, and a new approach that accounts for hydrodynamic characteristics is welcome to standardize results according to these factors. Future research should strive to reduce the risk of bias by (i) attending to a balance between female and male swimmers, looking for better sample homogeneity; (ii) providing important personal characteristics; (iii) controlling the evaluation conditions; (iv) providing the software validity and reliability; (v) blinding the outcome evaluators; (vi) providing data on the inter-evaluator reliability of outcome measurement or measures of error for the methodologies used (when applicable); (vii) providing information about whether swimmers are part of larger samples; and (viii) pre-registering the research protocols.

## Figures and Tables

**Figure 1 bioengineering-10-00308-f001:**
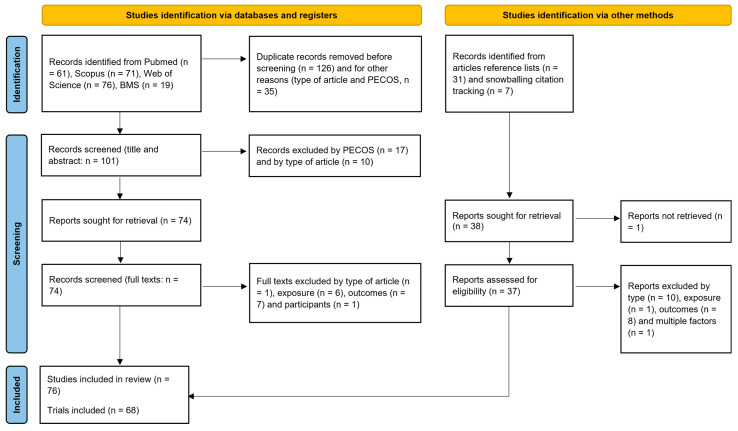
Search and screening processes used in the current study displayed as a PRISMA 2020 flow diagram.

**Figure 2 bioengineering-10-00308-f002:**
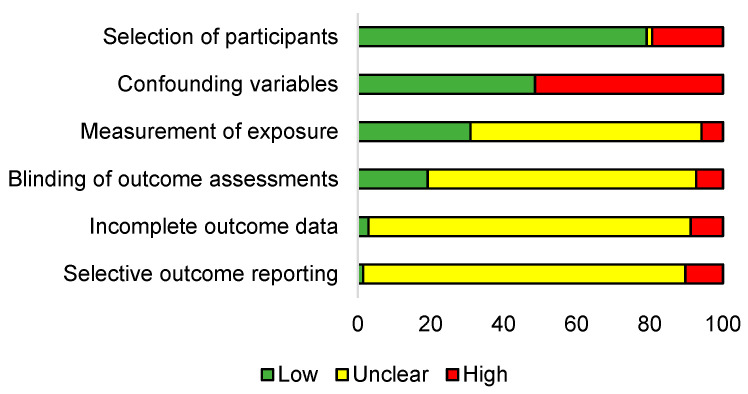
Percentage for each risk-of-bias domain regarding the included trials.

**Table 1 bioengineering-10-00308-t001:** Full search strategies for PubMed, Scopus, Web of Science databases, and Biomechanics and Medicine in Swimming Symposia.

Database	Observations	Search Strategy
PubMed	Nothing to report	(((((((swim*[Title/Abstract]) AND (intracycl*[Title/Abstract])) OR (“intra-cycl*”[Title/Abstract])) OR (IVV[Title/Abstract])) AND (velocity[Title/Abstract])) OR(speed*[Title/Abstract])) OR (accelera*[Title/Abstract])) OR (quick*[Title/Abstract])
Scopus	The search for title and abstract also includes keywords	((swim*[Title/Abstract]) AND (intracycl*[Title/Abstract] OR “intra-cycl*”[Title/Abstract] OR IVV[Title/Abstract])) AND (velocity[Title/Abstract] OR speed*[Title/Abstract] OR accelera*[Title/Abstract] OR quick*[Title/Abstract])
Web of Science	Title/abstract is not available in this database. The option “Topic” includes title, abstract and keywords, and was used instead	swim* (Topic) AND intracycl* OR “intra-cycl*” OR IVV (Topic) AND velocity OR speed* OR accelera* Or quick* (Topic)
Biomechanics and Medicine in Swimming Symposia	Title/abstract was not available in this database. The option “All Fields” was used instead	(All Fields:swim*) AND (All Fields:intracycl* OR “intra-cycl*” OR IVV) AND (All Fields:velocity OR speed* OR accelera* OR quick*)

**Table 2 bioengineering-10-00308-t002:** List of included trials and respective main characteristics (including the evaluated swimming technique, the participants characteristics, the used methodology, the conflicts of interests and the corresponding funding).

Study	Swimming Technique	Participants	Assessment	Protocol	Conflicting Interest and Funding
Miyashita [7]	Front crawl	Eight highly trained male and 1 sedentary female swimmers	Cable speedometer	100 m at best effort	Unreported
Holmer [10]	Breaststroke, front crawl	1 elite, 1 trained and 1 recreational swimmer	Accelerometer	1–2 min each at several different velocities up to their maximal velocity	Unreported
Craig, et al. [118]	Breaststroke	Twelve trained male swimmers (19 years)	Cable speedometer	5 repeated swims using a range of 20–30 upper limbs cycles per minute for the slowest swims up to his maximal velocity (50–60 stroke rate)	Unreported
Loetz, et al. [104]	Backstroke, breaststroke, butterfly, front crawl	1 male and 8 female swimmers	Electrical impedance	Sprint	Unreported
Manley and Atha [6]	Breaststroke	4 highly trained male and 4 trained female swimmers (14–16 years)	Swimming tachometer	12 m maximum, 12 m 50% maximum and 12 m acceleration from 50 to 100%	Unreported
Ungerechts [111]	Breaststroke	13 male and 9 female highly trained swimmers (14.5–20.5 years)	3D	Unreported	Unreported
Vilas-Boas [35]	Breaststroke	Thirteen highly trained male swimmers (15.8 ± 2.2 years)	Photo-optical method	3 × 200 m: 2 at submaximal velocities, 1 maximal effort	Unreported
Colman, et al. [126]	Breaststroke	25 male (19.9 ± 2.6) and 20 female (17.9 ± 3.07) elite swimmers	2D	25 m at 100 m competitive pace	Unreported
D’Acquisto and Costill [94]	Breaststroke	7 male (19.7 ± 1.5) and 8 female (19.0 ± 1.1 years) trained swimmers	Cable speedometer, 2D	Two all-out 15 yards (22.86 m)	Unreported
Alberty, et al. [26] Alberty, et al. [27]	Front crawl	Seventeen highly trained swimmers (21 ± 3 years)	Cable speedometer	2 × 25 submaximal with 4 x 50 m max in between to induce fatigue and 25 m front crawl test at maximal velocity 30 min before 200 and just after 200 m	Unreported
Barbosa, et al. [19]	Butterfly	Seven highly trained and elite male swimmers (18.4 ± 1.9 years)	3D	3 sets of 3 × 25 m as fast as possible	Unreported
Kjendlie, et al. [110]	Front crawl	10 children (11.7 ± 0.8) and 13 adults (21.4 ± 3.7 years)	2D	4 × 25 m front crawl at submaximal velocities	Unreported
Takagi, et al. [127]	Breaststroke	46 male and 35 female world-class swimmers	2D	25 m of 50, 100 and 200 m breast	Unreported
Barbosa, et al. [34]	Butterfly	3 male (17.6 ± 2.9) and 2 female highly trained swimmers (15.0 ± 1.4 years)	3D	3 × 200 m butterfly: 2 submaximal (75 and 85%), one maximal	Unreported
Balonas, et al. [123]	Backstroke, breaststroke, butterfly, front crawl	Twelve elite male swimmers (19.8 ± 3.5 years)	3D	Test until exhaustion	Unreported
Barbosa, et al. [21]	Backstroke, breaststroke, butterfly, front crawl	12 male and 5 female elite swimmers	3D	Incremental set of n × 200 m	Unreported
Novais, et al. [105]	Breaststroke	2 male (17.0 ± 0.0) and 2 female elite swimmers (17.5 ± 2.1)	3D	Incremental set of n × 200 m	Unreported
Schnitzler, et al. [22] Schnitzler, et al. [33]	Front crawl	6 male (22.3 ± 4) and 6 female (21.0 ± 2.4 years) elite swimmers	2D	5 × 25 m at paces of 3000, 400, 200, 100 and 50 m	Unreported
Tella, et al. [108]	Front crawl	10 male and 7 female highly trained swimmers (between 14–16 years)	Cable speedometer	2 × 25 m and 100 m at maximum velocity	Unreported
Leblanc, et al. [36]	Breaststroke	9 elite male (19.9 ± 2.3) and 9 trained swimmers (15.1 ± 0.9 years)	Cable speedometer	3 × 25 m trials at 200, 100 and 50 m race pace	Unreported
Barbosa, et al. [113]	Butterfly	Ten international male swimmers (18.4 ± 1.9 years)	3D	2 × 25 m at high velocity	Unreported
Tella, et al. [112]	Front crawl	Sixteen trained and highly trained swimmers (17.0 ± 0.8 years)	Accelerometer	2 × 25 m front crawl sprint	No conflicts of interest Funded by University of Valencia (UV-AE-20041029)
Figueiredo, et al. [16]	Front crawl	Eight highly trained male swimmers (20.3 ± 2.8 years)	3D	25 m near maximum	Unreported conflicting interest Funded by Portuguese Science and Technology Foundation (SFRH/BD/38462/2007)
Psycharakis and Sanders [20]	Front crawl	Ten highly trained and elite male swimmers (16.9 ± 1.2 years)	3D	One maximum swim	Unreported conflicting interest Funded by Greek State’s Scholarship Foundation
Arellano, et al. [106]	Front crawl	5 male and 8 female trained and highly trained swimmers (19.6 ± 2.2 years)	Cable speedometer	25 m as fast as possible	Unreported conflicting interest Funded by Secretary of State for Research, Ministry of Science and Innovation. Ref. DEP2009-08411. University of Granada, Physical Education and Sports Department and Research Group of Physical Activity and Sports on Aquatic Environment [CTS.527]
Psycharakis, et al. [37]	Front crawl	Eleven junior and senior elite and highly trained swimmers (16.9 ± 1.2 years)	3D	200 m race pace	Unreported
Schnitzler, et al. [32]	Front crawl	10 elite/highly trained swimmers (22.5 ± 3.6) and 12 trained swimmers (23.0 ± 1.7 years)	Cable speedometer	Four swim trials at 100, 80–90, 70–80 and 60–70%	Unreported
De Jesus, et al. [128]	Butterfly	Seven trained female swimmers (17.6 ± 2.0 years)	2D	2 × 100 m butterfly swim: one at submaximal and one at maximal velocity	Unreported
Fernandes, et al. [15]	Front crawl	Sixteen highly trained and trained swimmers (29.2 ± 10.3 years)	3D	Intermittent protocol with increments of 0.05 m/s each step and 30 s rest intervals	Unreported conflicting interest Funded by PTDC/DES/101224/2008 [FCOMP-01-0124-FEDER-009577]
Ferreira, et al. [120]	Front crawl	Nine male highly trained swimmers (18.0 ± 2.3 years)	3D	200 and 400 m race pace	Unreported conflicts of interest Funded by Portuguese Science and Technology Foundation [POCI/DES/58362/2004]
Figueiredo, et al. [90] Figueiredo, et al. [4] Figueiredo, et al. [91] Figueiredo, et al. [25] Figueiredo, et al. [30]	Front crawl	Ten male highly trained swimmers (21.6 ± 2.4 years)	3D	200 m race pace	No conflicts of interest Funded by Portuguese Science and Technology Foundation [SFRH/BD/38462/2007] and [PTDC/DES/101224/2008—FCOMP-01-0124-FEDER-009577]
Barbosa, et al. [3]	Backstroke, breaststroke, butterfly, front crawl	23 male and 22 female highly trained and trained swimmers (12.8 ± 1.2 years)	Cable speedometer	Maximal 4 × 25 m	Unreported
Feitosa, et al. [87]	Breaststroke, butterfly	12 male (14.4 ± 1.2) and 11 female highly trained and trained swimmers (12.7 ± 0.8 years)	Cable speedometer	Maximal 2 × 25 m	Unreported
Gourgoulis, et al. [31] Gourgoulis, et al. [93]	Front crawl	Nine female highly trained swimmers (18.4 ± 4.9 years)	3D	25 m trials at different paces	No conflicts of interest No funding
Morais, et al. [117]	Front crawl	62 male (12.8 ± 0.7) and 64 female highly trained and trained swimmers (12.0 ± 0.9 years)	Cable speedometer	3 × 25 m	Unreported conflicting interest Funded by Portuguese Science and Technology Foundation (SFRH/BD/76287/2011)
Figueiredo, et al. [92]	Front crawl	Thirteen trained swimmers (27.8 ± 10.9 years)	3D	30 min	Unreported
Komar, et al. [95]	Breaststroke	11 male and 7 female elite (20.8 ± 2.1) and recreational swimmers (20.4 ± 1.5 years)	3D	2 × 25 at maximal velocity + 4 × 25 m: 2 at 90 and 2 at 70% of the maximal velocity	Unreported conflicting interest Funded by CPER/GRR1880 Logistic Transport and Information Treatment 2007–2013
Matsuda, et al. [23]	Front crawl	7 elite (20.9 ± 0.9) and 9 highly trained swimmers (20.2 ± 1.6 years)	2D	30 m front crawl at 4 velocities: maximal velocity (Vmax) and 75, 85, and 95% Vmax	Unreported
Seifert, et al. [102]	Breaststroke	Seven highly trained swimmers (17.5 ± 2.2 years)	3D	3 × 200 m at 70% of their breast 200 m personal best	No conflicts of interest Unreported funding
Soares, et al. [107]	Front crawl	15 male (18.8 ± 2.4) and 13 female (16.5 ± 2.4 years) trained swimmers	Cable speedometer	50 m all-out	Unreported
Sanders, et al. [89]	Breaststroke	Two male elite swimmers (18 years)	3D	S1: 4 × 25 m front crawl maximal sprint. S2: 4 × 50 m front and back sprints	No conflicts of interest Unreported funding
Barbosa, et al. [85]	Backstroke, breaststroke, butterfly, front crawl	34 male (17.1 ± 4.1) and 34 female elite swimmers (15.0 ± 3.0 years)	Cable speedometer	Maximal 4 × 25 m	No conflicts of interest Funded by NIE acrf grant (RI11/13TB)
Dadashi, et al. [24]	Front crawl	13 and 5 female swimmers, 9 highly trained (19.3 ± 1.8) and 9 trained swimmers (16.0 ± 1.8 years)	Accelerometer	3 × 300 m at 70, 80 and 90% of their front-crawl 400 m personal best time with 6 min rest between trials	No conflicts of interest Unreported funding
De Jesus, et al. [28]	Front crawl	Ten male highly trained swimmers (19.8 ± 4.3 years)	3D	Intermittent incremental protocol of 7 × 200 m with increments of 0.05 m/s and 30 s resting intervals between steps	No conflicts of interest Funded by PTDC/DES/101224/2008 (FCOMP-01-0124-FEDER-009577) and CAPES /543110-7/2011
Figueiredo, et al. [116]	Front crawl	51 male and 52 female highly trained and trained swimmers (11.8 ± 0.8 years)	2D	25 m front crawl at a 50 m front crawl race pace	Unreported
Morais, et al. [98]	Front crawl	12 male (13.6 ± 0.7) and 15 female (13.2 ± 0.9 years) highly trained and trained swimmers	Cable speedometer	Maximal 3 × 25 m	Unreported
Seifert, et al. [101]	Front crawl	Five male elite swimmers (20.8 ± 3.2 years)	Cable speedometer	Three front crawl variants (with steps of 200, 300 and 400 m distances) incremental step test until exhaustion (with a 48 h rest period in-between)	Unreported Funded by PTDC/DES/101224/2008 (FCOMP-01-0124-FEDER-009577), CAPES/543110-7/2011 and Séneca Foundation 19615/EE/14.
Barbosa, et al. [114]	Backstroke, breaststroke, butterfly, front crawl	21 male and 4 female elite (15.7 ± 1.5), 11 male and 14 female highly trained (15.7 ± 3.6) and 18 male and 7 female recreational swimmers (22.9 ± 3.4 years)	Cable speedometer	Maximal 4 × 25 m	Unreported conflicts of interest Funded by NIE acrf grant (RI11/13TB)
Costa, et al. [109]	Backstroke, front crawl	Sixteen recreational swimmers (19.8 ± 1.1 years)	Cable speedometer	2 × 25 m	Unreported
Van Houwelingen, et al. [103]	Breaststroke	14 male and 12 female (20.0 ± 3.3 years) highly trained swimmers	2D	10 × 50 m (70% of the maximal velocity)	No conflicts of interest Funded by Stichting voor de Technische Weteschappen, grant number 12868
Barbosa, et al. [84]	Front crawl	12 male and 12 female recreational swimmers (22.4 ± 1.7 years)	Cable speedometer	All out 25 m freestyle pre (rest) and post (fatigue) test	No conflicts of interest Funded by NIE acrf grant (RI11/13TB)
Bartolomeu, et al. [122]	Backstroke, breaststroke, butterfly, front crawl	24 male and 25 highly trained and trained swimmers (14.2 ± 1.7 years)	Cable speedometer	Maximal 4 × 25 m	No conflicts of interest Funded by European Regional Development Fund [POCI-01-0145-FEDER-006969]; Portuguese Science and Technology Foundation [UID/DTP/04045/2013]
Gonjo, et al. [42]	Backstroke, front crawl	Ten male highly trained swimmers (17.5 ± 1.0 years)	3D	300 m at VO2 steady state	No conflicts of interest Funded by YAMAHA Motor Foundation for Sports (YMFS) International Sport Scholarship
Gourgoulis, et al. [17]	Breaststroke	Nine male trained swimmers (21.6 ± 4.2 years)	3D	25 m at maximal intensity	No conflicts of interest No funding
Morouço, et al. [88]	Front crawl	Nine male recreational swimmers (42.5 ± 9.5 years)	Cable speedometer	25 m at maximal intensity	Unreported conflicts of interest Funded by Portuguese Science and Technology Foundation (pest-OE/EME/UI4044/2013).
Morouço, et al. [99]	Front crawl	Twenty-two male highly trained swimmers (18.6 ± 2.4 years)	Cable speedometer	50 m time-trial	Unreported conflicts of interest Funded by Portuguese Science and Technology Foundation (UID/Multi/04044/2013)
Krylov, et al. [96]	Front crawl	Nine male elite swimmers (18.0–24.0 years)	2D	3 × 25 m self-selected pace at 100, 200 and 1500 m	No conflicts of interest Unreported funding
Silva, et al. [39] Silva, et al. [29]	Front crawl	Twenty-three male and 26 female trained swimmers (15.7 ± 0.8 and 14.5 ± 0.8 years)	3D	50 m at maximal velocity	No conflicts of interest Funded by Portuguese Science and Technology Foundation (SFRH/BD/87780/2012)
Correia, et al. [119]	Front crawl	Fourteen trained male swimmers (23.0 ± 5.0 years)	3D	200 simulating 400 m	No conflicts of interest Unreported funding
dos Santos, et al. [121]	Front crawl	Twenty trained swimmers (18.5 ± 3.9 years)	2D	Repeated 50 m maximum performance with 10 s interval	Unreported conflicts of interest No funding
Ruiz-Navarro, et al. [100]	Front crawl	Sixteen male trained swimmers (19.6 ± 3.3 years)	Cable speedometer	25, 50 and 100 m	Unreported conflicts of interest Funded by Ministry of Economy, Industry and Competitiveness (Spanish Agency of Research) and the European Regional Development Fund (ERDF); DEP2014-59707-P. Spanish Ministry of Education, Culture and Sport: FPU17/ 02761
Barbosa, et al. [124]	Front crawl	Fourteen male elite swimmers (25.7 ± 6.4 years)	Cable speedometer	25 m maximal sprint	No conflicts of interest Funded by Swedish Research Council
Barbosa, et al. [125]	Butterfly	One world-class male swimmer (26 years)	Cable speedometer	25 m maximal sprint	No conflicts of interest No funding
Engel, et al. [86]	Breaststroke	4 male (16 ± 0.7) and 6 female trained swimmers (14.9 ± 0.9 years)	Accelerometer	100 m moderate intensity	No conflicts of interest Funded by Federal Institute for Sports Science (ZMVI4-070804/19-21)
Morais, et al. [97]	Butterfly	10 male (15.4 ± 0.2) and 10 female (14.4 ± 0.2 years) highly trained swimmers	Cable speedometer	Three all outs	No conflicts of interest Funded by Portuguese Foundation for Science and Technology (UIDB/DTP/04045/2020)
Neiva, et al. [115]	Front crawl	16 male and 6 female recreational swimmers (39.9 ± 6.1 years)	Cable speedometer	2 × 25 m at maximal velocity	No conflicts of interest Funded by Portuguese Foundation for Science and Technology (UIDB04045/2020)
Fernandes, et al. [40]	Backstroke	12 male and 9 female swimmers, 16 elite (16.2 ± 1.0) and 15 trained (15.7 ± 1.3 years)	Cable speedometer	25 m at maximal velocity	No conflicts of interest Funded by Portuguese Science and Technology Foundation (2020.06799.BD)
Fernandes, et al. [41]	Front crawl	10 male (16.2 ± 1.8) and 17 female elite swimmers (18.3 ± 3.5 years)	Cable speedometer	25 m at maximal velocity	No conflicts of interest Funded by Portuguese Science and Technology Foundation (DFA/BD/6799/2020)

**Table 3 bioengineering-10-00308-t003:** Mean ± SD or median ± IQR mean velocity and IVV values obtained in the swimming trials included in the current study.

Swimming Technique	Competitive Level	Sprint	Extreme	Severe	Heavy	Moderate	Low
Backstroke	World class	-	-	-	-	-	-
	Elite	1.54 ± 0.11 m/s 13.18 ± 3.67%	-	1.29 ± 0.09 m/s 18.49 ± 2.44%	-	-	-
	Highly trained	1.19 ± 0.1 m/s 11.02 ± 4.17%	-	-	-	-	-
	Highly trained/trained	1.11 ± 0.63 m/s 6.99 ± 2.77%	-	-	-	-	-
	Recreational	0.96 ± 0.16 m/s 12.99 ± 4.94%	-	-	-	-	-
Breaststroke	World class	-	-	-	-	-	-
	Elite	1.23 ± 0.11 m/s 39.72 ± 4.47% 0.76 ± 0.18 m/s	-	1.04 ± 0.09 m/s 20.75 ± 4.8%	-	-	-
	Highly trained	1.35 ± 0.11 m/s 26.93 ± 3.38% 1.46 ± 0.33 m/s	-	-	-	-	0.92 ± 0.08 m/s 1.18 ± 0.22%
	Highly trained/trained	0.94 ± 0.11 m/s 45.34 ± 3.25%	-	-	-	-	-
	Recreational	0.81 ± 0.07 m/s 41.19 ± 6.69% 0.75 ± 0.20 m/s	-	-	-	-	-
Butterfly	World class	1.78 m/s 24.32%	-	-	-	-	-
	Elite	1.75 ± 0.09 m/s 21.86 ± 4.33%	-	1.21 ± 0.12 m/s 29.71 ± 7.54%	-	-	1.03–1.48 m/s 39.20 ± 11.50%
	Highly trained	1.15 (1.06–1.34) m/s 25.68 ± 14.72%	-	-	-	-	-
	Highly trained/trained	1.06 ± 0.16 m/s 26.98 ± 9.69%	-	-	-	-	-
	Trained	1.31 ± 0.10 m/s 27.87 ± 14.68%	1.29 ± 1.31 m/s 19.92 ± 22.48%	-	-	-	-
	Recreational	32.44 ± 6.92%	-	-	-	-	-
	World Class	-	-	-	-	-	-
Front crawl	Elite	1.84 ± 0.06 m/s 12.30 ± 2.39%	1.52 ± 0.11 m/s 5.23 ± 1.77%	1.43 ± 0.54 m/s 11.76 ± 4.01%	1.53 ± 0.12 m/s 9.70 ± 3.49%	12%	1.28 ± 0.11 m/s 6.87 ± 2.91%
	Elite/highly trained	1.80 ± 0.10 m/s 14.30 ± 2.40%	-	1.60 ± 0.10 m/s 14.10 ± 1.80%	1.40 ± 0.20 m/s 14.50 ± 1.60%	-	1.20 ± 0.20 m/s 14.30 ± 2.10%
	Highly trained	1.51 ± 0.16 m/s 6.99 ± 2.18%	1.74 ± 0.06 m/s 2.44 ± 0.74%	1.43 ± 0.13 m/s 8.62 ± 1.60%	1.40 ± 0.05 m/s 4.51 ± 0.2%	1.08 ± 0.06 m/s 0.17 ± 0.01%	1.11 ± 1.13 m/s 8.74 ± 15.67%
	Highly trained/trained	1.41 ± 0.14 m/s 5.24 ± 1.77%	-	1.06 ± 0.29 m/s 22 ± 6.50%	-	-	-
	Trained	1.36 ± 0.20 m/s 8.36 ± 2.28%	1.50 ± 0.08 m/s 9.20 ± 1.27%	1.30 ± 0.14 m/s 13.73 ± 2.89%	1.16 ± 0.11 m/s 9.25 ± 1.67%	1.06 ± 0.14 m/s 23 ± 5%	0.94 ± 0.76 m/s 15.83 ± 8.94%
	Recreational	1.28 ± 0.19 m/s 2.42 ± 0.78%	-	-	-	-	-

Legend: IVV quantified by dv/v is presented in the breaststroke row.

## Data Availability

Not applicable.

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
