# Peer review of "Intracycle Velocity Variation in Swimming: A Systematic Scoping Review"

_bioengineering, 2023, doi:10.3390/bioengineering10030308_

Round 1
Reviewer 1 Report
This interesting review discusses intracycle velocity variation within a swimming stroke cycle and its variation with technique, exercise intensity, energy expenditure, and fatigue as a relevant parameter for assessing the biomechanical and coordinative development of swimming technique.
Author Response
Thank you for your words and effort when reviewing our manuscript.
Reviewer 2 Report
The current systematic scoping review protocol was designed according to PRISMA 2020 and Prisma-ScR guidelines, as well as Cochrane recommendations and focused on the swimming IVV assessment that is retrospectively available for almost a century.
The manuscript is well written and present interesting findings.
I suggest that it is suitable for publication in its present form.
Author Response
Thank you for understanding the effort that we took when idealizing and writing this manuscript.
Reviewer 3 Report
It was a pleasure to review this systematic scoping review article. The authors should be congratulated for their efforts. The necessity of this review was well expressed. I hope that my comments might help to improve the quality of the manuscript, in particular in its clarity in a few sections.
Line 27 - Please change the keywords already presented in the title.
Line 30-33 – I suggest starting the introduction with the definition of IVV, followed by information about its evaluation and evolution in scientific history. Moreover, please insert the publishing years of the mentioned studies about the topic (here and after) through the explanation of the research history.
Line 57-59 – please provide references for this statement.
Line 89-102 – In my opinion, the explanation concerning the inclusion of this source could be included in section “2.2 information sources” rather than “2.1 eligibility criteria” in which any source utilized is presented.
Moreover remember that symposium is singular and for the plural you should use symposia (as it derives from Latin). Please change accordingly
Secondly you state that you considered articles published in any language. How so?
Lines 96-98 You state that the proceedings of the symposia are more valuable than some of the high impact peer review journal publications. This statement is pretty strong. I would suggest to modify. .
Line 112: UP to July 2022
Line 137: This sentence needs to be rephrased. It is too long and I personally have difficulty understanding it for this reason I am not able to suggest something different.
Line 146-148, Tables 2 and 3– The nomenclature utilized to classify participants appears to be different between text, Tables and original work by McKay et al. Please homologate or explain this discrepancy. Furthermore, it might be better to utilize the swimming-specific caliber framework recently published by Ruiz-Navarro et al (https://doi.org/10.1080/17461391.2022.2046174) instead of the general one presented by McKay et al., if possible.
Moreover, sedentary individuals should not be contemplated in this review (as you chose to select only competitive athletes) Maybe also children should be excluded?
Line 164-168 – I don’t find this supplementary material. Can you provide it for the peer-review process?
Table 2 – i) Why report information regarding conflict of interest and funding of each study? ii) format the first row with the headers. More, possibly insert the headers on each following page.
Line 202: Considering that no data WERE provided (data is plural)
Line 253 – it might be useful to include a summary table of all the information presented in section 3.3
Line 254 – Please indicate that the “evidence gap map” is presented as Supplementary material and “trials risk of bias evaluation” as Figure 2 (if correct).
Line 257-264 - For clarity, I suggest inserting the % value after each named group instead of as long (confusing) lists.
Line 260 – what is the difference between “youth”, “junior” and “young”? Moreover, why “and” between both “youth and Junior” and “young and master”?
Line 261 and following: I am confused here as how the intensity zones were detmined (same comment for line 291)
Line 332 – Substitute the “good-level counterparts” that might confound the reader.
Table 3 – Although the table represents a perfect summary of the data collected, a little more effort should be to improve its clarity. In particular: i) the utilization of both “m/s”, “%” divided by “and” more times could create confusion for the reader, also for the non-homogeneous formatting of each line (the attention of the reader should be focalized more on % representing IVV, the variable studied in the manuscript); ii) check the table, sometimes the measurement unit is missing; iii) The name of the swimming style is presented with a different formatting, iv) the classification of the swimmers is not clear and not in agreement with what is expressed in the text (see comment on line 146-148); v) format the headers of the table; vi) the information provided in the note should be presented and discussed (at least once) also into the main text of the paper; vii) evaluate the possibility to present the data also by graphs; viii) although you reported in lines 256-257 that most of the studied swimmers were males, it should be reasonable to not present speed data of males and females together because of the different speeds reached based on sex. Moreover, in table 3 it is hard to understand how the different intensity zones were defined and it is puzzling that the at moderate velocity is higher than at maximal. Or that highly trained have a higher speed than elite at the same intensity. I think this table needs more explanation both in the legend and in the text and the authors should find a way to make more easily understandable to readers.
Line 314 – eliminate the extra space
Line 350 – “…gained ground until today” poor expression, please change
Line 358 – “…eventual differences between female and male swimmers…” Please provide information (in the intro and/or discussion) about the possible differences found between males and females in the literature. Moreover, did you perform an analysis of IVV results based on sex (thus comparing IVV in males vs females)? Why not?
Line 370 and following: Better level swimmers present higher IVV however later in the paragraph it seems as though you are stating the opposite. Please clarify
Line 392-393 – did you perform an analysis of IVV results based on each method to estimate IVV (thus grouping IVV data collected by each method)? Why not?
Line 400; I would suggest you write “it is widely accepted” rather than very accepted.
Line 429 – I found the conclusions section very poor, without a summary of the results found, its explanation, and possible future perspectives. I suggest revisiting this section in its entirety, and also trying to add the “Future Directions” section as suggested by the journal’s guidelines for review studies.
Reviewer 4 Report
General Comments
· The topic is interesting and could help practitioners with the concept of intracycle velocity variation which is crucial for swimming technique.
· The authors have a solid base of methods and study procedures to determine study objectives.
· The writing quality is very well.
· Methods section is comprehensive, and reviewing style (PRISMA) is also strong.
· Tables are very nice and comprehensive.
I have a few edits to consider:
Abstract
· Please try to better explain the originality and practical reasons of the study before study aims.
· Please, use different keywords than were used in title.
Introduction
· The last sentence of the first paragraph seems so general and strong. Please move this sentence into the last paragraph of introduction.
· Try to use shorter sentences.
Methods
· Please check mistyping in Table 1.
· Table 3 could be rebuilt. I have some doubts about its necessarity. It could be better to more focus on the front crawl.
Discussion
· The discussion is a bit long with redundant information. I recommend the authors simplify this part so that the readers would benefit from the outcomes of this study.
· Please add “limits of the study” subtitle before the conclusions.
· Please check references according to the journal style.
Author Response
General Comments
- The topic is interesting and could help practitioners with the concept of intracycle velocity variation which is crucial for swimming technique.
- The authors have a solid base of methods and study procedures to determine study objectives.
- The writing quality is very well.
- Methods section is comprehensive, and reviewing style (PRISMA) is also strong.
- Tables are very nice and comprehensive.
Thank you for reviewing our manuscript and for your kind words.
I have a few edits to consider:
Abstract
- Please try to better explain the originality and practical reasons of the study before study aims. ??
- Please, use different keywords than were used in title.
We changed accordingly.
Introduction
- The last sentence of the first paragraph seems so general and strong. Please move this sentence into the last paragraph of introduction.
Thank you. Since the last sentence of the first paragraph refers to the study that is being described, we decided not to move it.
- Try to use shorter sentences.
We did our best to improve the writing style.
Methods
- Please check mistyping in Table 1. The table was checked.
- Table 3 could be rebuilt. I have some doubts about its necessarity. It could be better to more focus on the front crawl. Since swimming includes, both for teaching, training and competing purposes, four conventional techniques, we have focused on them all in our review. If we center our attention only in front crawl, we will be limiting the research performed in the current topic. We hope that you understand our intention of grouping all the most well-known swimming techniques.
Discussion
- The discussion is a bit long with redundant information. I recommend the authors simplify this part so that the readers would benefit from the outcomes of this study.
- Please add “limits of the study” subtitle before the conclusions.
- Please check references according to the journal style.
Thank you, we did as suggested.